# Effectiveness of a Safe Sex Education Module in Improving Condom Use among People Living with HIV: A Randomised Controlled Trial

**DOI:** 10.3390/ijerph191610004

**Published:** 2022-08-13

**Authors:** Azline Abdilah, Hayati Kadir, Kulanthayan Mani, Ganesh Muthiah

**Affiliations:** 1Department of Community Health, Faculty of Medicine and Health Sciences, Universiti Putra Malaysia, Serdang 43400, Selangor, Malaysia; 2Malaysian Research Institute of Ageing (MyAgeing), Universiti Putra Malaysia, Serdang 43400, Selangor, Malaysia

**Keywords:** education, HIV, condom use, safe sex, self-efficacy

## Abstract

The Human Immunodeficiency Virus (HIV) epidemic in Malaysia has transitioned to occurring through more sexual transmission than injecting drugs in 2018. According to reports, the increase was caused by poor condom compliance and a lack of health programmes to prevent sexually transmitted infections (STIs) among people living with HIV (PLWH). The purpose of the study was to create, implement, and evaluate the impact of a safe sex education module on condom use among PLWH. A single-blinded, parallel randomised controlled trial was conducted at Seremban district. The intervention group received additional health information geared toward safe sex education based on Social Cognitive Theory (SCT). The study primary analysis was the intention to treat, and the overall effects of the intervention were assessed using a generalised linear mixed model (GLMM). There was no significant difference between groups in terms of sociodemographics, sexual history, mean condom usage frequency score, or STI incidence at the study baseline. Receiving the module was linked to increased condom usage frequency (β = 1.228, % CI = 0.850, 1.606). When compared to conventional treatment provided in Seremban health clinics, this module effectively increases condom usage frequency among PLWH.

## 1. Introduction

In 2018, the global Human Immunodeficiency Virus (HIV) infection epidemic revealed that 37.9 million people had been infected with the virus [1]. In Malaysia, an estimated 87,041 people were living with HIV (PLWH) in 2018. According to a national survey, the HIV epidemic in Malaysia has transitioned from injecting drugs to sexual transmission [2]. In 2018, the proportion of new HIV infections acquired through sexual transmission has increased up to 90% of the annual total [3].

Condom use has proven to be an effective preventive measure in reducing HIV transmission and limiting the spread of the disease in areas where the pandemic has concentrated in key populations [4]. Condom use is an essential part of a comprehensive and long-term strategy for preventing HIV and other sexually transmitted infections (STIs) among PLWH. When condoms are used consistently, they offer the most significant preventive benefit [5].

Previously, most countries’ HIV preventive efforts, including Malaysia’s, focused primarily on HIV-negative people with high-risk sexual behaviour. Unfortunately, the sexual risk behaviour of PLWH was not given enough attention. The implications of inconsistent condom use by PLWH will exacerbate the HIV infection epidemic and increase the risk of reinfection with drug-resistant virus strains among those PLWH in treatment [1]. According to a study conducted in seven Asian countries, the prevalence of inconsistent condom usage with multiple sexual partners, whether regular or casual partners of unknown or other HIV status, is significant (43–46%) among PLWH [6].

Sex workers, people who inject drugs (PWID), men who have sex with men (MSM), and transgender people were also more likely to report irregular condom usage [4,6,7]. Furthermore, previous research has linked inconsistent condom usage among PLWH with HIV-positive sexual partners, having less information about HIV or STIs, having poor negotiation skills, and perceiving that condom use reduces enjoyment [8,9,10,11].

Given these concerns, a health education intervention based on health theories that emphasise the cognitive pathways involved in behaviour change is required to prevent HIV transmission. Social Cognitive Theory (SCT) is one of the most widely used models of sexual transmission risk behaviours [12]. In brief, this model shows that individuals go through a cognitive process of weighing the pros and cons of practising safer sex. This cognitive process involves considering knowledge about HIV, expectancies related to using condoms, and social norms which may influence an individual’s self-efficacy.

SCT-framed interventions are thought to improve condom use and reduce sexual risk behaviour by improving individuals’ behavioural skills and perceptions of their ability to use condoms (self-efficacy). Several studies have demonstrated that SCT-framed interventions are successful at improving condom use and reducing STI incidence [13,14,15].

Despite the challenges mentioned earlier, in Malaysia, up to date, no studies have been conducted to assess the effectiveness of a theory-based sex education module in improving PLWH’s self-efficacy towards better condom compliance. The module developed in this study could be adopted and added into the routine health education given to PLWH during their regular treatment care visit in health clinics. The application of this module would lead to improvements in their compliance with preventive measures, subsequently reducing the disease’s burden and its complications.

## 2. Materials and Methods

### 2.1. Trial Registration

First Registration Site: Thai Clinical Trials Registry URL: http://www.thaiclinicaltrials.org/show/TCTR2020021100 (accessed on 6 February 2022).

### 2.2. Subject Recruitment and the Study Protocol

Data collection was conducted from December 2020 to February 2021. The researchers conducted a single-blind, individual, randomised, controlled trial (RCT) with pre-and post-intervention evaluations. In a 1:1 ratio, the participants were randomly assigned to one of two arms: intervention or control. The intervention responder received an additional safe sex education module, while the control respondent received usual care. The study was conducted in government health clinics in Negeri Sembilan’s Seremban District. Seremban is the most populous of the seven districts in Negeri Sembilan, located in southern Malaysia. The Seremban Health District administers eight health facilities that provide care to PLWH.

Anyone diagnosed with HIV who was eligible with the study’s criteria and registered in government health clinics in the Seremban District was included in the study. The sample size (n) in this study was calculated by using the two population means formulae [16]. The estimation of the sample size in this study was based on a study done by Olley et al. in 2006 in Abuja, Nigeria. The study concluded that participants who underwent individual health education intervention effectively increase safe sexual practices among PLWH. Preliminary data indicated that the mean frequency of condom use of the intervention group is 8.45. If the mean of the control group is 6.4, we needed to study 42 samples per group to reject the null hypothesis, with a probability (power) of 0.8. The Type I error probability associated with this test of this null hypothesis is 0.05. An independent *t*-test statistic was used to evaluate this null hypothesis. With an additional 20% dropout rate, the total sample size was 100.

By handing out flyers outlining the study, those potential respondents were asked for screening consent. The questionnaire was provided to all PLWH patients. The HIV/AIDS unit at the Seremban District Health Office then created a list of all PLWH individuals eligible for treatment. After then, each person was assigned a number. A simple random sample was given out using a table of random integers created by the software. A random sequence was generated in this investigation using computer software from the website http://www.sealedenvelope.com. (accessed on 15 December 2020) The PLWH person who was randomly assigned to the control or intervention arm was the randomisation unit. Restricted block randomisation was utilised as the method of randomisation. A block of two was chosen to ensure the equilibrium of each arm. The single blinding strategy was adopted in this study due to financial and technical restrictions. Each participant’s intervention assignment was unknown to them, but it was known to the researcher and others involved in delivering the intervention module. All responders were blinded at all sites 14 (See Figure 1).

### 2.3. Structured Intervention Program

The study duration took nearly three years to be completed, as it was started from December 2018 to May 2021. The intervention was built around a Malay and English-language module. The training module for medical officers and paramedics was developed using an educational intervention model derived from the SCT model. The validation of the module training manual was based on pre- and post-test scoring with a questionnaire by a group of specialists, including public health and family medicine physicians.

The intervention’s sessions were guided by a trained person from the training module. The activities were handled through an interactive session, allowing for listening, questions, and answers. Areas covered during the session included information on HIV transmission and prevention, safe sexual practices, condom use, and individual risk assessment, followed by risk reduction strategies. The trainees evaluated the participants’ self-efficacy to change sexual transmission risk behaviour or practising safe sex with their intimate partner by encouraging and suggesting the steps of risk reduction. The intervention group received a total of 3 intervention sessions consisting of few health seminars, brainstorming, open discussion, and role-play. The intervention took place over a 1-day course, and each session lasted for less than 1 h. Thus, the total duration of the intervention was 3 h. It was only done once during the baseline.

On the other hand, the control group received standard care that included regular counselling on ARV adherence by a physician and other necessary health advice based on patient active complaint/s, if indicated at the baseline. The data collection was conducted for both groups initially at the baseline, then at one month and three months of follow up.

### 2.4. Outcome Measures

The results of the study were acquired using a reliable and validated self-administered questionnaire. The questionnaire consists of four sections which include Section 1 (sociodemographic characteristics of the respondents), 2 (socioeconomic characteristics of the respondents), 3 (sexual history), and 4 (social cognitive theory (SCT) constructs on condom use). In relation to condom use, the respondents had to score their condom usage frequency from ‘0’ to ‘4’ for the frequency of condom use at the last sex act, since a higher score indicates better condom compliance. For instance, ‘0’ means ‘Never’ and ‘4’ means ‘Always’.

### 2.5. Statistical Methods

The study data was analysed using IBM Statistical Package for Social Science (SPSS) version 25 (IBM, Chicago, United States) In this study, descriptive and inferential statistics were used. Descriptive statistics were employed to describe the characteristics of the respondents. The 2 Chi-squared Test for categorical variables and the U Mann-Whitney test for continuous variables compared respondents in the intervention and control groups. A generalised linear mixed model (GLMM) was used to examine the overall effects of group, time, and group–time interaction effects on the mean of self-efficacy on condom use, while adjusting for variables. The confidence interval for mean estimations was set at 95%. The significance level, alpha α, was set at 0.05. The primary result of this study was the change in the mean of self-efficacy on condom use from baseline to endpoint. When compared to per-protocol analysis, intention-to-treat analysis was used, covering any loss to follow-up or dropout cases.

### 2.6. Ethical Approval and Consent to Participate

The Medical Research Ethics Committee (MREC) of the Ministry of Health (MOH) and the Ethics Committee for Research Involving Medical Research and Human Subjects at the University Putra Malaysia provided ethical approval for this study (JKEUPM). On respondent data sheets, the identification number was used instead of patient identifiers. Because the data was consolidated into a database, the respondents did not share the personal study data. On the other hand, respondents could write to the researchers and request access to the study findings. There are no actual or potential conflicts of interest in this study.

## 3. Results

### 3.1. Responses Rate

The flow chart in Figure 1 shows the participant recruitment procedure. One hundred PLWH were finally selected to participate in the study, with 50 in each of the two groups. All of the participants in the intervention (100%) and control groups (100%) had attended their respective health education and scheduled standard care sessions. The response rates for the intervention and control groups at 1-month follow-up and 3 month follow-ups were 90.0 and 86.0%, respectively Overall, there were 10.0 and 14.0% dropouts from the intervention and control groups, respectively.

### 3.2. Missing Data Analysis

The main outcome analysis in this study was intention-to-treat analysis. There is a total of 3.8% of missing data identified during data analysis. The missing data were missing completely at random (MCAR) and handled using multiple imputations.

### 3.3. Comparison of Baseline Characteristics of the Participants

Table 1 shows the sociodemographic and sexual history characteristics of participants. The control group’s median age was 31.00 (14) and 33.5 (13) in the intervention group. Besides, most participants in both groups were male, Malay, had a secondary education level, and were employed. Moreover, for the sexual history characteristic, there was no difference in sexual orientation, sexual partners, multiple partners, or condom use between the two groups. As a result, both groups were comparable at the baseline.

Table 2 shows the descriptive result of changes in the mean score of condom use frequency between the control and intervention groups. There were significant differences in condom use among participants at both the first month and third months, as the intervention group showed an increase in median (IQR) from 3.00 (2) baseline to 5.00 (1) at one month and 5.00 (2) at three months of post-intervention.

### 3.4. Main Effects of the Intervention

To examine intervention effects while controlling for covariates such as participants’ age, education, household income, sexual orientation, multiple partners, and baseline data, a GLMM was used. On the other hand, the combination of variables produced the best model, as evidenced by the lowest Akaike information criterion (AIC) and Bayesian Information Criterion (BIC). Sensitivity analysis was carried out using both intentions to treat (ITT) and per-protocol analysis (PPA). Table 3 reveals the condom use frequency score of a participant in the intervention group to be 1.228 points higher than a participant in the control group (β = 1.228, t = 6.371, SE = 0.193, *p* < 0.001, 95% CI = 0.850, 1.606). There is a significant difference in condom use frequency score at 3 month follow up as the participants scored 11.2% higher than at baseline. Besides, those participants in the intervention scored 1.377 points and 1.5 points higher at 3 months and 1 months as compared to baseline, respectively. Moreover, Table 4 shows that the effects of group, time, and group–time interaction on condom use frequency were all significant. These indicate that intervention improved the condom use frequency of participants, especially at the first-month follow-up; however these rates slightly dropped at the third-month follow-up. However, despite not receiving the intervention, the control group still showed improvements in condom use frequency during the follow-ups. See Figure 2.

### 3.5. Sensitivity Analysis

The effects of the group’s condom use frequency remained significant even after the GLMM analysis was performed using per-protocol analysis, as shown in Table 5. 

## 4. Discussion

The number of participants recruited in this study was adequate, as it accounted for the total number of the study sample size with an additional 20% dropout. During baseline data collection, all 100 participants turned up. The proportion of participants who completed all two-point follow up was 88%. The attrition rate was slightly higher in the control group (14%) than in the intervention group (10%). Attrition was due to uncontactable participants, participants who were no longer interested in continuing the study, difficulties obtaining work leave from their employers, and transferring to other districts or state health facilities for a continuum of care. The researcher approached them several times more, but they were still unable to elicit a response.

The participants’ baseline sociodemographic characteristics show that the median age ranged from 31 to 33 years old. The majority of both groups were male (70–82%) and of Malay ethnicity (70–72%). The distribution of participant characteristics is similar to that found in national reports, reflecting Malaysia’s PLWH composition [2,18]. According to Suleiman et al., in 2019, PLWH in Malaysia are predominantly male (86.2%), and more than 70% of new HIV infections in 2018 were reported among people aged 20 to 39 years old. Furthermore, the study’s use of sexual history characteristics, such as the mode of sexual transmission, reveal that most of the participants are men who have sex with men (MSM), accounting for 60% of the total. This percentage matched the national report, as MSM account for most HIV diagnoses (61.1%). There is no significant difference in the sociodemographic features of respondents in the control and intervention groups, according to a sub-analysis. Because the findings of this study may assist people who have a similar feature, the baseline results have some research implications. As a result, the necessary action can be implemented. The baseline attributes association also aims to minimise selection bias through randomisation.

The primary outcome of this module was to increase the condom use frequency of the respondent while having sexual intercourse with their sexual partners. Post-intervention analysis shows significant differences between groups. The improvement markedly increased from baseline to the first-month follow-up but slightly increased at the third month.

Few interventional studies have measured condom use; however, the studies used a wide range of measures and showed mixed results [7,8,19,20,21]. For instance, a study done in Nigeria found similar results when compared with the control group [19]. According to a health education intervention based on SCT, which was conducted among 120 PLWH in Southern Africa, a general increase in the frequency of condom use at three-month follow-up was found. A proportion of 82% of the respondents at follow-up reported always using condoms versus 33% at baseline. However, an ANCOVA model showed no significant difference between the intervention and control conditions on this measure [8].

The overall effect on improving condom usage in this study shows that the intervention has increased condom use frequency among respondents, notably at the first-month follow-up but slightly decreased at the third-month follow-up. Furthermore, the control group shows minor improvement between baseline and the third-month follow-up. These changes were more significant in the intervention group than the control group due to the considerable improvement of self-efficacy on condom use among intervention participants. Few studies have found a link between condom use self-efficacy and condom use frequency, with a higher score of condoms use self-efficacy among sexually active people who routinely used a condom at their previous sex act [22,23,24]. Sensitivity analysis of ITT and PPA results showed significant improvements, indicating that the intervention had a favourable impact on condom use frequency.

## 5. Conclusions

The study reveals that the intervention has improved condom use frequency among the respondents. The control group still has slight improvement from the baseline and the third-month follow-up despite not receiving the intervention. There are few recommendations for future research. Some of the theory components and additional variables, such as knowledge and awareness, can be measured in future studies. The respondents may obtain a better understanding of HIV transmission and safer sex practices to prevent recurrent STIs and improve condom use frequency. Secondly, given the positive effects of the health intervention module, it is recommended for the Seremban District Health Office organisation to add-on this module in standard care while managing treatment care among PLWH. It is recommended that at least a further session be delivered two to three months after the initial session as a re-enforcement. According to the findings, self-reinforcement, practising correct methods, and overcoming barriers to condom use prevent STIs.

## Figures and Tables

**Figure 1 ijerph-19-10004-f001:**
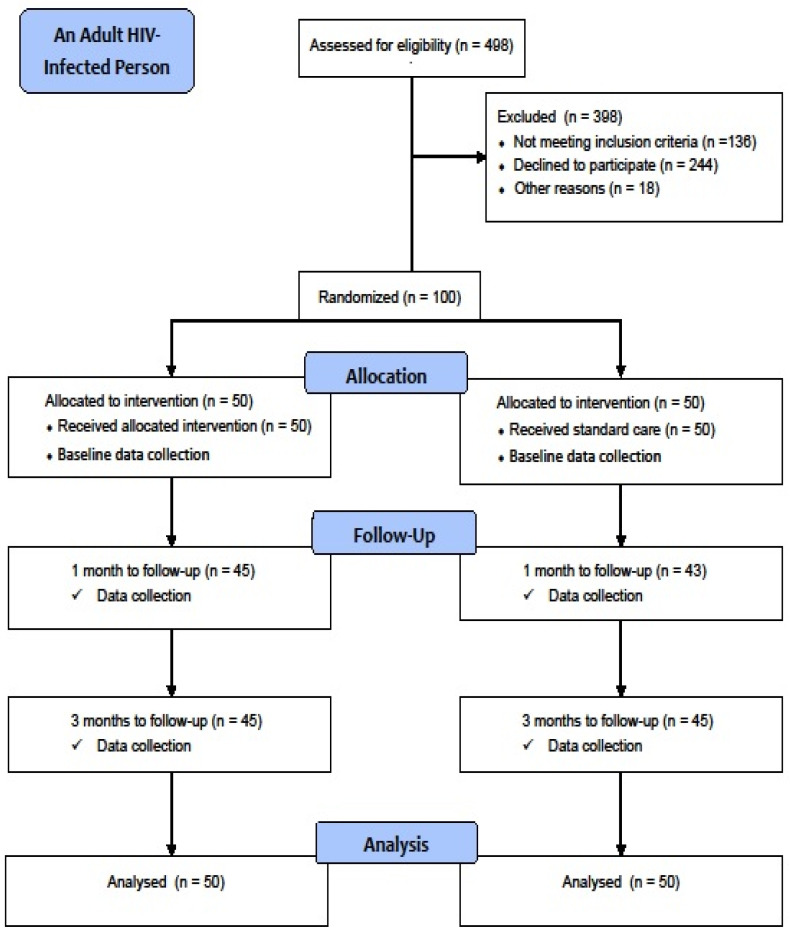
Flow chart diagram of the study using CONSORT statement [17].

**Figure 2 ijerph-19-10004-f002:**
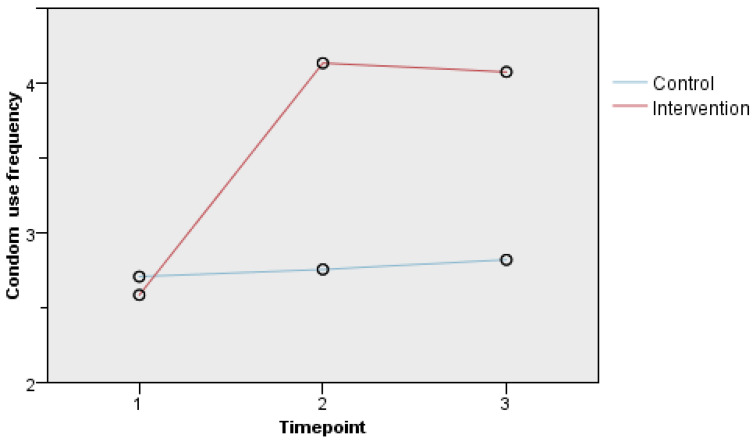
Interaction Plot Between Group and Time Point for Condom Use Frequency (ITT).

**Table 1 ijerph-19-10004-t001:** Baseline sociodemographic and sexual characteristics (*n* = 100).

	Group Median (IQR)/*n* (%)		Statistical Test *p*-Value	
	Control	Intervention		
Age (years)	31.00 (14)	33.5 (13)	1096.50 ^φ^	0.29
Household Income (RM)	2500 (2800)	1850 (1800)	1046.50 ^φ^	0.16
Gender				
Male	41 (82.0)	35 (70.0)	2.07 ^β^	0.35
Female	6 (12.0)	9 (18.0)		
Transgender	3 (6.0)	6 (12.0)		
Ethnicity				
Malay	36 (72.0)	36 (70.0)	1.00 ^β^	0.91
Chinese	5 (10.0)	3 (6.0)		
Indian	6 (12.0)	8 (16.0)		
Others	3 (6.0)	4 (12.0)		
Educational Status				
No formal	0 (0)	0 (0)	0.73 ^β^	0.61
Primary	4 (8.0)	4 (8.0)		
Secondary	26 (52.0)	30 (60.0)		
Tertiary	20 (40.0)	16 (32.0)		
Employment Status				
Non-Employed	6 (12.0)	8 (16.0)	0.33 ^β^	0.77
Employed	44 (88.0)	42 (84.0)		
Mode of Sexual Transmission				
Heterosexual Gay Bisexual	16 (32.0) 27 (54.0) 7 (14.0)	24 (48.7) 21 (42.0) 5 (10.0)	2.68 ^β^	0.26
Sexual Partner/s				
Spouse or Lovers	21 (54.0)	23 (46.0)	2.09 ^β^	0.35
Casual Partner	24 (48.0)	18 (36.0)		
Sex workers or paid for sex	5 (10.0)	9 (18.0)		
Multiple Partners				
No	17 (34.0)	25 (50.0)	2.63 ^β^	0.11
Yes	33 (66.0)	25 (50.0)		
Condom use frequency	3.00 (2.0)	3.00 (2.0)	1122.50 ^φ^	0.36
STI No Yes	20 (40.0) 30 (60.0)	19 (38.0) 31 (63.0)	0.042 ^φ^	0.84

^β^ χ2 Chi-squared Test, ^φ^ U Mann-Whitney Test, STI: Sexual Transmission Infections.

**Table 2 ijerph-19-10004-t002:** Between-group difference of total scores of condom use frequency.

	Group Median (IQR)		
	Baseline	1-Month	3-Months
The total score of Condom use Control	3.00 (2)	3.00 (2)	3.00 (2)
Intervention	3.00 (2)	5.00 (1)	5.00 (2)
Man-Whitney U Test	1122.50	426.00	468.50
*p*-value	0.361	<0.001	<0.001

**Table 3 ijerph-19-10004-t003:** Fixed coefficient of variables for condom use frequency (ITT).

Variable	Coefficients	Std. Error	t	*p*-Value	95%CI	
					Lower	Upper
Group						
Intervention	1.228	0.193	6.371	<0.001 *	0.850	1.606
Control	1					
Time						
3-month	0.112	0.041	2.746	0.006 *	0.032	0.192
1-month	0.048	0.041	1.180	0.238	−0.032	0.128
Baseline	1					
Time × Group						
3-month × Intervention	1.377	0.058	23.865	<0.001 *	1.264	1.490
1-month × Intervention	1.500	0.058	26.056	<0.001 *	1.387	1.613
Baseline × Intervention	1					

Random Effect: Mean estimate = 0.864, SE = 0.130, Z-Value = 6.638, *p* < 0.001. * Significant at *p* < 0.05.

**Table 4 ijerph-19-10004-t004:** Fixed effects of group, time, and group–time interaction on total condom use frequency scores (ITT).

Total Scores of Condom Use	Parameter	F	df1	df2	*p*-Value
Participant					
	Group	18.206	1	1773	<0.001 *
	Time	514.275	2	1773	<0.001 *
	Group × Time	418.832	2	1773	<0.001 *

Using generalised linear mixed model adjusted for covariates with multiple imputated data. * Significant at *p* ≤ 0.05.

**Table 5 ijerph-19-10004-t005:** Comparison of fixed coefficients for the group, with and without.

Variable	Intention to Treat		Per-Protocol Analysis		Coefficient Difference	Percentage Coefficient Difference
	Coefficient	Sig.	Coefficient	Sig.		
Condom use frequency						
Intervention	1.228	<0.001	1.133	<0.001 *	0.095	+7.74
Control	1		1			

* Significant at *p* ≤ 0.05.

## Data Availability

Not applicable.

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
