# Peer review of "Effectiveness of a Safe Sex Education Module in Improving Condom Use among People Living with HIV: A Randomised Controlled Trial"

_ijerph, 2022, doi:10.3390/ijerph191610004_

Round 1

Reviewer 1 Report

Dear Authors, thank you for submitting your manuscript. I had the chance to read it and based on the issues highlighted below, I believe that it is not ready for a publication. I hope you'll find my comments helpful. 

Please, use consistently persons living with HIV rather than HIV-positive people. Also, though minor point, the acronym used is PLWH rather than PLHIV. Please use men who have sex with men instead of men-sex men. Lines 53-54:  PLHIV acronym has been already introduced earlier. There is no need to spell it again.

I have difficult time understanding these sentences: The important aspect of this concept is an individual's self-efficacy in deciding whether or not to use a condom when having sex. According to this concept, individuals who go through a cognitive process capable of balancing the benefits and drawbacks of practising safer sex may have an impact on their 63 self-efficacy.

First, I there is no concept introduced in the previous sentence. I am not sure if the Authors refer to the model rather than concept. Second, it sounds like self-efficacy is a key component of the model. Third, it seems like processing the benefits and drawbacks of using condoms impacts self-efficacy. I believe it is important to explain in what ways. Intuitively, one may think that perceiving more benefits than drawbacks can increase self-efficacy. However, I am not familiar with studies that have shown these effects. You may want to consider expanding a bit more and add more citations.

Readers may not be familiar with the concept of self-efficacy. It would be helpful to introduce it briefly, explaining also how it can be improved. Also, it is unclear if the Authors refer to general self-efficacy or self-efficacy related to condom use.

Please, elaborate more on the procedure to determine the sample size. It is quite unclear if a formal power analysis was conducted.

Lines 93-97 are a repetition of a previous paragraph.

These two sentences have the same meaning, you may want to choose one or the other. The significance level, alpha α, was set at 0.05. The p-value of the decision rule was less than 0.05.

Akaike corrected information criterion (ACIC) is usually referred to as Akaike information criterion (AIC)

Please double check these results, t=0.193, SE=6.371, P<0.001. If the B coefficient is 1.228 and the SE is 6.371 it is impossible that the p value is <0.001. It looks more like t=6.371 and SE 0.193 (as reported in a table). I have some concerns regarding the use of linear approach to condom use, which is a count variable. I believe it would be more appropriate to use a poisson model.

Figure 2 looks a bit off. The y axis labels are 2,3,3,4,4,5… ??

Overall, the manuscript needs a lot of work to improve clarity, consistency and provide more information on the rationale for the specific intervention. Data imputation was not mentioned in the manuscript, unless I missed it. However, it looks like missing data were imputed. Please provide more information on the imputation. I would suggest using a different analytic approach. It is counterintuitive to use linear models while non-parametric approach was used to perform between group comparisons. I believe that a poisson model may be better suited for the specific data.

Author Response

Dear Dr./ Mr./Ms. (Reviewer),

Thank you for giving me the opportunity to submit a revised draft of my manuscript. I appreciate the time and effort that you have dedicated to providing your valuable feedback on my manuscript.  I have been able to incorporate changes to reflect most of the suggestions provided by the reviewer. I have highlighted the changes within the manuscript with red colour. Here is a point-by-point response to the reviewers’ comments and concerns. Please see the attachment.

Reviewer 2 Report

This manuscript used a randomized controlled trial based on Social Cognitive Theory (SCT) and described the effectiveness of a sex education module in improving the Condom use among PLWH in Malaysia, where HIV transmission has been transitioned from injecting drugs to sexual.

Specific questions/comments

  1. In Figure 1, 45 individuals received intervention and 43 received standard care, what is the explanation for analysis 50? For 1 month follow-up, what is the explanation for only carried out intervention in intervention group, but no standard care in control group
  2. Line 182, the intervention participants should be 90% (45/50), not 94.44%
  3. How did you get the result in line 185
  4. Line 201-205 are replication from line 193-197
  5. In figure 2, what is the time point units? Is it Months?

Author Response

Dr./ Mr./Ms. (Reviewer),

Thank you for giving me the opportunity to submit a revised draft of my manuscript. I appreciate the time and effort that you have dedicated to providing your valuable feedback on my manuscript.  I have been able to incorporate changes to reflect most of the suggestions provided by the reviewer. I have highlighted the changes within the manuscript with red colour. Here is a point-by-point response to the reviewer’s comments and concerns. Please see the attachment.

Round 2

Reviewer 1 Report

Thank you for addressing some of my concerns. I noticed that the authors are still using a linear approach to analyzing count data. Overall your responses are satisfactory. Please, be consistent with the use of PLWH.

For example Line 45 -- Please replace HIV-positive people with PLWH 

Also

lines 60-61 --- The cognitive process is as by considering knowledge... Seems like something is missing between is and as or as and by.... 

Author Response

We thank the reviewer for pointing out the errors that we are overlooked. We have corrected the error as you have suggested. Moreover, in line 60-61, we have changed to appropriate sentences. Thank you.